# Efferocytosis and Respiratory Disease

**DOI:** 10.3390/ijms241914871

**Published:** 2023-10-03

**Authors:** Wenxue Zheng, Zhengjie Zhou, Xiaoping Guo, Xu Zuo, Jiaqi Zhang, Yiming An, Haoyu Zheng, Yuan Yue, Guoqiang Wang, Fang Wang

**Affiliations:** The Key Laboratory of Pathobiology, Ministry of Education, College of Basic Medical Sciences, Jilin University, Changchun 130021, China; zhengwx22@mails.jlu.edu.cn (W.Z.); zhengjie21@mails.jlu.edu.cn (Z.Z.); guoxp21@mails.jlu.edu.cn (X.G.); zuoxu21@mails.jlu.edu.cn (X.Z.); zhangjiaqi22@mails.jlu.edu.cn (J.Z.); anym22@mails.jlu.edu.cn (Y.A.); zhenghy22@mails.jlu.edu.cn (H.Z.); yueyuan@jlu.edu.cn (Y.Y.)

**Keywords:** efferocytosis, apoptosis, macrophage, respiratory disease

## Abstract

Cells are the smallest units that make up living organisms, which constantly undergo the processes of proliferation, differentiation, senescence and death. Dead cells need to be removed in time to maintain the homeostasis of the organism and keep it healthy. This process is called efferocytosis. If the process fails, this may cause different types of diseases. More and more evidence suggests that a faulty efferocytosis process is closely related to the pathological processes of respiratory diseases. In this review, we will first introduce the process and the related mechanisms of efferocytosis of the macrophage. Secondly, we will propose some methods that can regulate the function of efferocytosis at different stages of the process. Next, we will discuss the role of efferocytosis in different lung diseases and the related treatment approaches. Finally, we will summarize the drugs that have been applied in clinical practice that can act upon efferocytosis, in order to provide new ideas for the treatment of lung diseases.

## 1. Introduction

Under homeostasis, cells, as the smallest functional unit of the body, go through a regular process of dying and being replaced by new cells [1]. There are various forms of cell death, such as apoptosis, necrosis, pyroptosis and ferroptosis, with apoptosis predominating in the steady state [2,3]. Dead cells need to be promptly cleared to form new structures, maintain organ function and prevent secondary necrosis. Otherwise, the contents of dead cells are released into the surrounding environment, damaging surrounding cells and tissues, which may cause inflammation and autoimmune diseases [4].

The timely clearance of dying and dead cells by phagocytes is crucial for maintaining host homeostasis. In most tissues, professional and non-professional phagocytes exist to clear cell corpses, a process known as efferocytosis. Professional phagocytes are mostly tissue-resident cells, such as macrophages and dendritic cells, whereas non-professional phagocytes are mostly adjacent cells, such as epithelial cells and fibroblasts [5], while macrophages are the main cells involved in efferocytosis (Figure 1) [4]. Effective efferocytosis can prevent the release of cellular contents, prevent secondary necrosis and inhibit inflammation. It can also promote the release of anti-inflammatory mediators, such as IL-10, to facilitate the resolution of inflammation [6].

## 2. Mechanisms of Efferocytosis

The phagocytosis of cellular corpses is mainly divided into four steps: (1) the smell” stage, which recruits phagocytes through the “find-me” signal; (2) the “taste” stage, which identifies dead cells through the “eat-me” and “don’t eat-me” signals; (3) the phagocytic stage, which ingests cellular corpses; (4) the digestion and degradation of dead cells [7] (Figure 2).

### 2.1. The “Smell” Stage

Apoptotic cells are capable of releasing various “find-me” signals to recruit phagocytes to the dead site [8], such as adenosine triphosphate (ATP) and uridine triphosphate (UTP), which are recognized by macrophage purinergic receptor P2Y2 [9,10], lyso-phosphatidylcholine (LPC) and sphingosine 1-phosphate (S1P), which can, respectively, bind to macrophage surface G protein-coupled receptors G2A and S1P1-5 [11,12,13], and chemokine CX3CL1 [14], which is released from apoptotic lymphocytes in a caspase and Bcl-2-dependent manner and mediates the macrophage recognition of apoptotic cells by interacting with macrophage CX3CR1 receptors through an unknown mechanism [8] (Table 1). CX3CL1 can also enhance efferocytosis by stimulating the expression of milk-fat globulin-EGF-factor 8 (MFG-E8), a phagocytic ligand, in phagocytes [10]. These signals have a dual effect; firstly, they can attract phagocytes to approach apoptotic cells; secondly, they can enhance the expression of phagocytic receptors by regulating the cytoskeleton of phagocytes [5], and some find-me signals may improve the efferocytotic function of non-professional phagocytes and play a role in immune regulation [15,16,17]. In addition, intercellular adhesion molecule 3 (ICAM3 or CD50) and thrombospondin 1 (TSP-1) on dying cells can, respectively, act upon CD14 and CD36 on macrophages, to promote the macrophage recognition of dying cells [18].

### 2.2. The “Taste” Stage

After phagocytic cells are recruited to the site of dying cells, it is necessary to distinguish between apoptotic cells and healthy cells through the “eat-me” signals expressed on the surface of dying cells for precise phagocytosis in the next step. Different changes occur on the surface of apoptotic cells, such as exposure of phosphatidylserine (PS) to the outer membrane lobules, epitope changes in the intracellular adhesion molecule-1 (ICAM1), exposure of calreticulin and changes in cell surface charge and glycosylation conformation [25]. Correspondingly, the “eat-me” signals that can be expressed on the surface of apoptotic cells include phosphatidylserine (PS), modified intracellular adhesion molecules, calreticulin, oxidized low-density lipoprotein, cell-bound thrombospondin (TSP), antibodies and complement opsonins [26] (Table 1).

The most effective, evolutionarily conserved and well-studied “eat-me” signal is PS exposed to the outer leaves of the plasma membrane of apoptotic cells [5]. In healthy cells, PS is kept on the inner leaves of the plasma membrane by two specific enzymes called flippase and scramble, which help maintain the asymmetry of PS on the cell membrane. When apoptosis occurs, these enzymes are cleaved by caspase 3, leading to the cleavage and inactivation of flippase and the activation of scramble; then, PS flips outward to the cell surface and is recognized by phagocytes [5,27,28]. Apoptotic cells can also expose their surface calreticulin, as an “eat-me” signal. For example, in cancer cells, the translocation of calreticulin from the endoplasmic reticulum to the plasma membrane occurs during apoptosis induction accompanied by endoplasmic reticulum stress, and then the CD91 (low-density lipoprotein receptor protein) mediates calreticulin recognition and subsequent phagocytosis [23,24].

The recognition of eat-me signals by phagocytes requires phagocytic receptors on the surface of phagocytes. There are two ways in this process, one is direct binding, and the other is indirect binding through bridging molecules. Phagocytic receptors such as brain-specific angio-genesis inhibitor 1 (BAI1), T cell immunoglobulin mucin receptor TIM1-TIM4 (TIM family) and Stabilin2 (Stab2) can directly bind to PS, while tyrosine kinase Tyro3/Axl/Mer (TAM) family receptors, scavenger receptor SCARF1, integrin receptor family complexes with CD36 or transglutaminase 2 (TG2) are bound to PS through soluble bridging molecules, such as Gas6 and protein S [22].

After BAI1 directly binds to PS through thrombospondin repeat (TSR), it recruits and activates the ELMO1-DOCK180 complex to initiate intracellular signaling, induce actin cytoskeleton rearrangement and promote efferocytosis [19]. The TIM receptor family is a type I cell surface glycoprotein that binds to PS through the N-terminal IgV domain and mediates the phagocytosis of apoptotic cells [10]. TIM-3 can promote the cross-presentation of apoptotic cell-associated antigens [29], while TIM-4 mediated phagocytosis can trigger a unique non-classical autophagy form called LC3-associated phagocytosis (LAP) [10,30]. It is worth noting that, although TIM-4 can directly interact with PS, it needs to cooperate with integrins or other receptors to transduce cell signals, and cannot be carried out independently [5]. Stabilin2 is a type I surface receptor that mediates the direct interaction with PS and triggers efferocytosis via adaptor protein GULP and thymosin β4, which regulates actin polymerization [20,31], while Stab2 can also promote binding to apoptotic cells in cultured cells [32].

The TAM receptor family is involved in a variety of physiological processes and has distinct but closely related roles in efferocytosis; all three receptors can mediate efferocytosis. The TAM receptor indirectly recognizes PS on apoptotic cells by binding the PS-binding serum protein Gas6 and ProS1, and subsequently dimerizes, activates Rac and promotes phagocytosis in apoptotic cells [10,21]. The integrin αvβ3 receptor family can also indirectly bind to PS, the integrin αv/β3/CD36 receptor complex binding to PS via TSP-1, while the integrin αv/β3/TG2 receptor complex interacts with PS via MFG-E8 [22]. Serum protein C1q can act as a bridging molecular by binding to the SCARF1 scavenger receptor on phagocytes, but also to apoptotic cell membrane-associated receptors A2 or A5 (AnnexinA2/A5) [33].

The reason why phagocytes recognize the “eat-me” signals is to accurately phagocytize apoptotic cells; in addition, phagocytes can also combine with the “don’t eat-me” signals on healthy cells through anti-phagocytosis receptors, and then activate phosphatase, mainly SH1/2, to prevent normal cells from being phagocytosed. Interestingly, when both the “eat-me” signals and the “don’t eat-me” signals are present on the cell, the “don’t eat-me” signals are superior to the “eat-me” signals, so phagocytosis does not occur [34].

The “don’t eat-me” signals that have been discovered so far mainly include CD47, CD24 and CD31 [5] (Table 1). CD47, which is expressed in normal cells but is changed or decreased in terms of expression in apoptotic cells, can activate the signal inhibitory regulatory protein-α (SIRP α) on macrophages, phosphorylate their cytoplasmic domains and activate phosphatase SHP1/2, thereby inhibiting phagocytosis [35]. CD24 is highly expressed on tumor cells and it binds to sialic acid-binding Ig like lectin 10 (Siglec10) to inhibit the clearance of tumor cells [36]. And CD31 has homologous interactions between healthy cells and macrophages, so it can also inhibit phagocytosis [37].

### 2.3. The Phagocytic Stage

The uptake of dead cells is mediated by the Rho family of small GTP enzymes, including RhoA, Rac, Cdc42, etc., which transition between resting, inactive GDP binding and active GTP binding states under the action of specific guanine nucleotide exchange factors (GEFs) [38,39]. RhoA activation can inhibit the phagocytosis of apoptotic cells, while Rac and Cdc42 can enhance phagocytosis [39]. Once the phagocytic cell recognizes the dying cell, the Rho family relocates on the plasma membrane and triggers a series of signaling pathways; then, the cytoskeleton rearranges through the dynamic network of actin fibers below the plasma membrane, the plasma membrane invaginates, local extravasation occurs [4] and a phagocytic cup is formed on the side facing the apoptotic cell [40], ultimately forming a phagosome. There are two main pathways involved in this process. One is the recruitment of the adapter protein GULP by binding LRP1 (CD91) to calreticulin or Stab2 to PS, thereby activating Rac [20,41]; the other is that, after the Mer tyrosine kinase (MerTK) or integrin receptor family indirectly binds to PS, engulfment and cell motility protein 1 (ELMO1) and dedicator of cytokinesis protein 1 (DOCK180) bind, and are then recruited to the cell membrane by the adapter protein CrkII, forming the ELMO1-DOCK180-CrkII complex [19,42,43], which further activates the RHO family, especially Rac1 [40]. Both of the above pathways promote phagocytosis by activating Rac, indicating that Rac plays a crucial role in the phagocytic process. Moreover, Cdc42 can promote phagocytosis by stimulating the formation of filopodia [44].

### 2.4. Digestion Phase

The completion of phagocytosis is not the end, but just the beginning. The phagosomes formed by phagocytosis need to fuse with lysosomes, which is also a multi-step process. The phagosomes need to mature first, then target lysosomes and fuse with lysosomes. This process is mainly mediated by the Rab GTP enzyme family, which, when activated, has functions such as transportation, vesicular fusion, Rab-related signaling and activation of other downstream Rab family members [45,46]. After completion of phagocytosis, Rab5 is recruited and activates early endosome antigen 1 (EEA1) and class III phosphatidylinositol 3-kinase VPS34 in an unknown way, which mediates the fusion of phagosomes with early endosomes, forming early phagosomes [44,47]. EEA1 endows vesicles with targeting from the plasma membrane to early endosomes, and there are two Rab5 binding sites on it, but neither site alone can mediate its stable binding with Rab, so synergistic action with phosphatidylinositol-3-phosphate (PI3P) is required [47]; VPS34 is rapidly recruited after the formation of early phagosomes, catalyzing the formation of PI3P by phosphatidylinositol, and further catalyzing the maturation of phagosomes [44].

Early phagosomes continue to transform into late phagosomes, with the most significant marker being the loss of Rab5 but the acquisition of Rab7 [44]. Rab7 is crucial for phagocytic maturation, acidification and fusion with lysosomes. The most critical proteins in this process are the Rab5 effector Mon1a/b and the homologous fusion and protein sorting (HOPS) complex [44]. When Mon1 still binds to Rab5, it can recruit the Rab7 binding protein Ccz-1 to promote binding to GDP-Rab7. At the same time, Mon1 can inactivate Rab5, and the Mon1/Ccz-1 complex plays a role in the activation process of Rab7 [48,49]. After binding to Rab7, the HOPS complex can serve as a GDI displacement factor to remove Rab7 from GDP dissociation inhibitors (GDIs) in order to activate Rab7 [50]. Rab7-interacting lysosomal protein (RILP) and oxysterol-binding protein-related protein 1 (ORPL1) directly combine with molecular motor motilin/motilin to mediate effective lysosomal fusion, which also requires complete microtubules [4].

After the fusion of late phagosomes and lysosomes, the newly formed phagolysosomes rely on proteases, nucleases and lipases in the lysosome to digest the phagocytic cell corpses [4]. The digested products are transformed and reused by macrophages or excreted from the body; otherwise, their accumulation can trigger inflammation and cause diseases [51].

## 3. Regulation of Efferocytosis

### 3.1. Approaches to Target the Taste Stage

Due to the fact that the taste stage is the recognition of apoptotic cells by macrophages, that is, the “eat-me” signal sent by apoptotic cells interacts with phagocytic receptors on macrophages, the regulation of phagocytic receptors may be an important way to improve efferocytosis. For example, IL-4 may enhance efferocytosis by increasing the expression of the phagocytic receptors Stabilin1 and Stabilin2 [52]. In a wild-type mouse model of colitis induced by the dextran sulfate sodium (DSS), BAI-1 mRNA levels were reduced in intestinal epithelial cells and colon tissue, while apoptotic cells were reduced in the colon epithelium of BAI-1 transgenic mice, and the pathology of colitis was also alleviated [53]. In cancer, reducing the number of anti-phagocytosis receptors allows for cancer cells to be engulfed, which may be a potential target for the treatment [5].

Many inhibitors of efferocytosis can exert their effects by blocking the recognition of apoptotic cells by macrophages. High mobility group box protein 1 (HMGB1) can bind to various macrophage receptors, for example, binding to the soluble receptor for advanced glycation end products (RAGE) can block macrophage recognition of PS, binding to integrin αV can block the interaction with the bridging molecule MFG-E8 [6,54,55,56]. RAGE and Annexin A5 can also downregulate efferocytosis by blocking the recognition of apoptotic cells by masking PS on apoptotic cells [56,57].

IL-4 and IL-13 induce the production of potential PPAR γ-activating ligands 13-hydroxyoctadecadienoic acid (13-HODE) and 15-hydroxyeicosatetraenoic acid (15-HETE) through STAT6 or 15-lipoxygenase to increase the expression and activity of PPAR γ, so as to increase the “alternative activation” of phagocyte surface receptors and the secretion of the bridging molecule adiponectin, enhance efferocytosis and inhibit inflammation [6,58,59,60]. IL-4 and IL-13 can also enhance PPAR δ of macrophages, promote the expression and release of the bridging molecules and make macrophages acquire anti-inflammatory function [6]. However, both of these receptors obtain these functions by combining with other nuclear receptors to form heterodimers, so liver × receptor (LXR) and retinoid × receptor α (RXRα) may play a role in the enhancement of efferocytosis by increasing the expression of MerTK [5], but whether IL-4 and IL-13 can directly interact with LXR and RXRα is unknown [6]. Although all LXR ligands can effectively enhance efferocytosis, some LXR agonists can increase liver triglyceride in some mouse models; therefore, developing new powerful and effective LXR agonists without adverse side effects may be beneficial to clinical use [22,61,62].

### 3.2. Approaches to Target the Phagocytic Stage

The phagocytic stage is mainly influenced by affecting the balance of Rac-1/RhoA. The enhancement of this stage mainly improves the ability of macrophages to phagocytose apoptotic cells by increasing the activity of Rac1, most of which are modified lipids. A modified phosphatidylserine, lyso-phosphatidylserine (LPS), is produced by the oxidation of the fatty acyl groups of PS in apoptotic neutrophils through the NADPH oxidase-dependent pathway [63,64]. LPS interacts with G2A receptors on macrophages to stimulate the production of prostaglandin E2 (PGE2), thereby enhancing the activity of cAMP and PKA-dependent Rac1 [6]. However, it should be noted that the effect of PGE2 on efferocytosis is concentration-dependent, with an enhanced efferocytosis effect within a concentration range of less than or equal to 1 nM, while an inhibitory effect is observed at doses greater than 10 nM [65]. Similar to LPS, lysophosphatidic acid (LPA) inhibits efferocytosis by activating RhoA [66]. Another molecule that can affect the Rac-1/RhoA balance is lipoprotein A4, which can activate Rac and RhoA at the same time, but the final result is more inclined toward the activation of Rac, which depends on protein kinase A [22,67].

When the mitochondrial oxidative stress level is elevated, the production of reactive oxygen species (ROS) also increases relatively, which activates the NOD-like receptor thermal protein domain-associated protein 3 (NLRP3), an inflammasome, and its related signaling pathways, ultimately leading to the inhibition of Rac1 expression at the gene level [68]. Mitochondrial uncoupling protein 2 (Ucp2) can regulate the NLRP3 signaling pathway by reducing the formation of ROS, so as to enhance the efferocytosis effect [69]. The efferocytosis effect can also be improved by reducing the membrane potential of macrophage mitochondria [18].

### 3.3. Approaches to Target the Digestion Stage

After the fusion of phagosomes and lysosomes, the cell corpse is digested, and the products obtained from digestion can affect the subsequent phagocytosis.

#### 3.3.1. Sugar

The deficiency of glycolytic enzyme 6-phosphofructose-2-kinase and fructose-2,6-bisphosphatase (PFKFB3) in macrophages can lead to a decrease in efferocytosis ability [70]. In addition, the reuse of glucose derived from apoptotic cells can serve as a part of the signaling pathway and/or provide energy to enhance subsequent efferocytosis [71]. That is to say, enhanced glycolysis in efferocytotic macrophages can promote actin aggregation and the sustained uptake of apoptotic cells.

#### 3.3.2. Lipid

Lipid metabolism induces the activation of PPAR γ and δ, LXR α and β and RXR α and β in macrophages, to promote efferocytosis (as previously mentioned) [72]. LXR signaling can regulate the expression of transglutaminase 2 (Tgm2), a co-receptor of αvβ3- integrin, and promote the formation of phagocytes [73,74]. LXR is also involved in Del-1-dependent efferocytosis and macrophage reprogramming to a pro-resolving phenotype [26,75]. In addition, the oxysterols produced by lipid metabolism are active inhibitors of the activation of the inflammasomes NLRP3, which promote subsequent efferocytosis [68,76].

#### 3.3.3. Amino Acid

Macrophages have been proved to utilize arginine and ornithine produced by the decomposition of apoptotic cells from the first round of phagocytosis to promote subsequent efferocytosis [5,77]. Yurdagul and colleagues found that, under the catalysis of inducible nitric oxide synthase (iNOS), arginine produces the precursor of polyamines, ornithine, in macrophages with a pro-catabolic phenotype. Ornithine is then converted into putrescine under the action of ornithine decarboxylase (ODC) [71,77,78]. Putrescine can enhance the mRNA stability of GTP exchange factor Dbl through the RNA binding protein HuR, which in turn upregulates and activates Rac1, leading to changes in the actin cytoskeleton and promoting the internalization of apoptotic cells [18,71,79,80].

#### 3.3.4. Transporter

Several members of the solute carrier (SLC) membrane transporter family, including glucose transport protein type 1 (GLUT1; encoded by gene Slc2a1) that promotes lactate release and monocarboxylate transporter 1 (MCT1; encoded by Slc16a1), are upregulated during efferocytosis [18]. SLC12A2 and SLC12A4 play different roles in the chloride-sensing pathway. SLC12A2 mediates the influx of Cl^−^, while SLC12A4 has the opposite effect. The influx of Cl^−^ plays a brake role in efferocytosis, so the inhibition or loss of SLC12A2 can lead to an enhanced efferocytosis [11,81]. By altering the activity or expression of selected solute carrier proteins (such as SLC2a1) while regulating the activity of cholesterol transporters ABCA1 and SLC16a1, efferocytosis can be promoted [5].

## 4. Efferocytosis and Respiratory Disease

In healthy individuals, apoptotic cells can rarely be detected in lung tissue [82], which may be due to the fact that normal airway macrophages can phagocytize and remove apoptotic cells in time. However, in acute or chronic lung inflammatory diseases, the rate of apoptosis is increased or the clearance of apoptotic cells is defective, which leads to the accumulation of apoptotic cells [83,84]. Experiments have proved that this pathological process is mainly due to the faulty efferocytosis of airway macrophages [85].

### 4.1. COPD

Chronic obstructive pulmonary disease (COPD) is a common chronic respiratory disease characterized by airflow restriction and airway inflammation. COPD is often caused by long-term exposure to toxic particles or gases, such as cigarette smoke, which is why the disease can be prevented and treated. The inflammatory characteristics of this disease have sparked a series of explorations into anti-inflammatory drugs, but most of them are limited to improving inflammation and alleviating respiratory symptoms and cannot stop the development of the disease. At present, there is a large amount of evidence indicating that there are defects in the efferocytotic function during the occurrence and development of COPD, leading to secondary necrosis and inflammation. The targets related to efferocytosis have also been extensively studied as new therapeutic directions for COPD.

LC3-related phagocytosis (LAP) is known to be essential for effective efferocytosis. Asare and others discovered the imbalance in the LAP pathway in COPD and cigarette smoke exposure for the first time. In the process of COPD, the expression of Rubicon, a key mediator in the lap pathway, decreases, leading to weakened efferocytosis and accumulation of apoptotic cells; subsequently, the release of inflammatory mediators exacerbates inflammation [86]. The downregulation of CD44, CD163, VSIG4 and MARCO in alveolar macrophages can also lead to their dysfunction, making it difficult for inflammation to subside [87,88]. The activation pathway of CR3 mediated by CD44 relies on divalent cations, especially Zn^2+^, but not Ca^2+^. This pathway is related to efferocytosis, so there is a deficiency in the efferocytosis of Zn-deficient alveolar macrophages [89,90].

The interaction between surfactant proteins A (SP-A) and SP-D and SIRP- α can inhibit efferocytosis, especially SP-D which is significantly reduced in COPD patients [82,91,92]. Cigarette smoke also damages the clearance of apoptotic cells through the oxidant-dependent activation of RhoA and the inhibition of Rac1, leading to actin polymerization defects commonly required for effective efferocytosis [93,94,95]. In addition, a reduction in PPAR-c, a decrease in methylation, an increase in ROCK or the colonization of microorganisms in the airway, especially *Hemophilus influenzae*, may lead to the defects in the efferocytotic function [96,97,98,99].

The damaged efferocytotic function is of great significance for the occurrence and development of COPD, so many studies have been conducted to achieve the goal of treating COPD by restoring efferocytotic function. Lovastatin can increase efferocytosis in various ways, such as inhibiting HMG-CoA reductase, altering the balance of membrane-bound RhoA and Rac-1, inhibiting oxidative stress and inhibiting matrix metalloproteinase-9 (MMP-9) [89]. Statins can also increase efferocytosis by increasing PPAR [66]. There is a positive correlation between available glutathione and phagocytosis, as Galactin-3 can induce macrophages to transform into a more “M2” phenotype through its interaction with CD98, increasing the phagocytic capacity and promoting the production of available glutathione [100,101]. Azithromycin and Rosiglitazone can also play a certain therapeutic role in COPD by increasing the phagocytosis of apoptotic cells by alveolar macrophages [98,101].

### 4.2. Asthma

Asthma is also a common heterogeneous chronic respiratory inflammatory disease, characterized by airway hyperresponsiveness and airway obstruction. Unlike COPD, the airway obstruction caused by asthma is reversible [102]. Efferocytosis can trigger a series of anti-inflammatory events, mediated by receptors such as MerTK, and executed by the production of interleukin-10 (IL-10) [103,104]. In asthma, there are fewer macrophages that produce IL-10, indicating that efferocytosis may be impaired [103,105]. Felton and others found that the enhanced efferocytosis of alveolar macrophages is helpful to solve allergic airway inflammation in mice, making the clearance of apoptotic cells a potential therapeutic strategy. This depends on MerTK, and enhancing Mer activity may be a specific mechanism of targeted efferocytosis [103,106].

In neutrophilic asthma, a decrease in the level of galactose agglutinin 3 (gal-3) results in the inability of apoptotic neutrophils to be completely cleared, leading to impaired efferocytosis and the persistence of inflammation. However, after the addition of exogenous gal-3, the ability of macrophages to clear apoptotic cells is restored. IL-1β, an inflammatory mediator in the airway of patients with neutrophilic asthma, is visibly increased, and its level is negatively correlated with gal-3, so adding exogenous gal-3 can inhibit the production of IL-1β and promote the resolution of inflammation [102]. Axl, an apoptotic cell recognition receptor only expressed in human airways or alveolar macrophages, was significantly reduced in asthma, and macrophages could not recognize apoptotic cells, so they could not be eliminated. The accumulated apoptotic cells suppressed the release of anti-inflammatory mediators such as IL-10 and TGF-β. And the shedding of Axl was also related to the damage of efferocytosis, while IFN could restore the expression of Axl on airway macrophages [107,108,109]. Research has shown that the expression of Neurofibroin-2 (Nrp2) is increased in neutrophil asthma mice and is a negative regulatory factor for neutrophil asthma. Therefore, it can regulate the efferocytosis of macrophages by promoting the degradation of endocytic apoptotic cells [109].

### 4.3. COVID-19

In recent years, the world has been plagued by the COVID-19 epidemic. Vaccination has greatly alleviated the impact of the epidemic in a short time, but the rapid mutation of the virus has brought great difficulties to the research and development of vaccines. Among the numerous variant strains, the Delta variant strain dominates, and people who have received vaccines can also become infected. Vaccines alone cannot control the epidemic; therefore, finding treatment methods that can reduce the severity of the disease seems more important [110]. The cause of this major public health and safety incident is severe acute respiratory syndrome coronavirus 2 (SARS-CoV-2), which mainly infects ciliated airway epithelial cells [111,112]. After infection, this virus damages the airway epithelium, alveolar epithelium, causing various clinical symptoms, mainly pneumonia, acute respiratory distress syndrome and multiple organ dysfunction [111,113,114,115]. SARS-CoV-2 infection can cause various lung cells to die in different ways, such as apoptosis, necrosis or pyroptosis [116]. At the same time, these apoptotic cells accumulate in large quantities due to impaired efferocytotic function, resulting in the production and release of pro-inflammatory cytokines. This may also be the main reason for various complications of COVID-19.

COVID-19 infection has been shown to hinder the normal function of alveolar macrophages. Recent studies have shown that the phagocytosis of SARS-CoV-2-AC reduces the transcription of PS receptors, including scavenger receptors CD36, SRA-1, integrin α V β5 (ITGB5) and T cell immunoglobulin mucin receptor 4 (TIM4), but the transcription of MerTK will not be decreased [111,117]. That is to say, the expression of receptors related to efferocytosis on dying cells infected with SARS-CoV-2 is inhibited, thereby impairing the sustained clearance of apoptotic cells by macrophages. It is speculated that SARS-CoV-2 infection can also damage the release of sphingosine-1-phosphate (S1P) involved in the process of efferocytosis, which can also slow down the process of eliminating apoptotic cells caused by SARS-CoV-2 infection [111,118].

In summary, targeting damaged efferocytosis may be a new therapeutic strategy against COVID-19. Gas6, a bridging molecule, can promote the combination of phosphatidylserine (PS) exposed on apoptotic cells with TAM receptors on macrophages, activating the receptors and triggering subsequent cellular signal transduction, promoting the phagocytosis of apoptotic cells, inhibiting innate immune responses in affected tissues, preventing inflammation and autoimmunity and protecting surrounding healthy tissues [10,119,120,121]. ADAM-17, a disintegrin and metalloprotein 17, can cleave phosphatidylserine receptors, and its deletion has been shown to increase efferocytosis in vivo and enhance the anti-inflammatory response [122,123,124]. The inhibition of ADAM-17 activity can reduce cytokine storm and the excessive aggregation of neutrophils [110,125]. Some immunomodulators used in COVID-19 may adjust the increased cytokine level by enhancing ADAM-17 activity [126]. Some compounds have been selected and have entered clinical trials for their ability to inhibit the function of ADAM-17, but they have been discontinued due to adverse events [110,127]. Therefore, drugs used to inhibit ADAM-17 need to be carefully selected.

### 4.4. Influenza

Influenza is caused by the influenza virus, which can be divided into three types: A, B and C. Among them, the influenza A virus (IAV) is the most serious, which can cause major public health and security incidents all over the world. IAV mainly replicates in respiratory epithelial cells. After the virus enters, it inhibits host cell protein synthesis and rapidly begins replication, causing clinical symptoms, ranging from upper respiratory tract infection to severe pneumonia [128]. At present, the prevention and treatment methods are very limited, and the main measure to be taken against it is vaccination. However, the antigenicity of the influenza A virus is prone to mutation and the effectiveness of influenza vaccines is not lasting, which further increases the difficulty of prevention and treatment. As a result, it has caused multiple global pandemics throughout history.

Pulmonary alveolar macrophages are key cells in controlling influenza, which can inhibit the replication of IAV and promote its clearance. They can also promote efferocytosis and the regression of lung inflammation. The increased efferocytosis of alveolar phagocytes can prevent fatal acute lung injury caused by lung infection, while the defect in efferocytosis of alveolar phagocytes can lead to chronic pneumonia [129,130]. Mannose-binding lectin (MBL) is an innate immune pattern recognition molecule existing in the lung that can bind to IAV. The downregulation of the tumor necrosis factor (TNF) release and reduced apoptosis mediated by MBL in human macrophages may represent its mechanism for promoting IAV clearance, reducing local tissue inflammation and oxidative damage and limiting collateral damage to the lungs through increased efferocytosis [130]. Angiotensin-(1-7) [Ang-(1-7)], a pro-apoptotic mediator, is able to increase the binding and phagocytosis of alveolar macrophages to apoptotic neutrophils in vitro, as was found by subjecting it to action in mice, and it can also enhance neutrophil apoptosis. The effects of Ang-(1-7) may be significantly beneficial in reducing lung injury and achieving better clinical outcomes after IAV infection [131]. The Bacillus Calmette–Guerin (BCG) vaccine is currently the only vaccine available for anti-tuberculosis, but its more important role is as an immunotherapeutic agent for treating other diseases, as it can provide non-specific protection for other diseases by increasing the recruitment and activation of macrophages [129]. Sanjay Mukherjee gave the BCG vaccine via nasal drops to mice, which showed a significant increase in the efferocytosis of alveolar phagocytes, and all of them survived after infection with the IAV, while all mice in the PBS-treated group died. After inserting apoptotic 5,6-carboxyfluorescein diacetate succinimidyl ester positive (CFSE^+^) mouse lung epithelial (MLE) cells into the alveoli of mice, the clearance efficiency of the BCG intervention group was significantly higher. BCG was also found to increase the expression of TIM4 on apoptotic cells, which is the first time that the role of TIM-4 in efferocytosis and the mechanism of BCG increasing efferocytosis were reported [129].

### 4.5. Other Lung Diseases

Acute respiratory distress syndrome (ARDS), the most severe form of acute lung injury (ALI), is characterized by alveolar injury, pulmonary edema and hypoxic respiratory failure caused by alveolar epithelial and endothelial cell death, alveolar neutrophil infiltration, etc. [132,133,134]. The pathogenesis of ALI/ARDS is not well understood, and the therapeutic approaches have limitations [133]. Fas/FasL of the tumor necrosis factor (TNF) family is known to promote apoptosis, and the inhibition of its associated signaling attenuates lung injury [133,135]. During the ARDS phase, neutrophils have an extended lifespan but eventually undergo apoptosis [136,137]. Therefore, the clearance of apoptotic cells by lung macrophages is crucial in attenuating lung injury. It has been found that the alveolar macrophage in ARDS patients does have an efferocytotic error, and IL-8 induces the onset of macrophage classical activation, which can inhibit efferocytosis; therefore, blocking IL-8 may make it possible to promote the clearance of apoptotic cells and reduce inflammation [138]. In addition, blocking high-mobility group box protein 1 (HMGB1) or promoting the activation of AMP-activated protein kinase (AMPK) can upregulate macrophage efferocytosis [139].

Cystic fibrosis (CF) is an autosomal recessive disorder capable of invading multiple organs, most commonly in the respiratory and digestive systems. The lungs are primarily characterized by impaired mucosal ciliary clearance and subsequent bacterial colonization [140], caused by mutations in the CF transmembrane regulatory protein (CFTR) in the lung epithelium that result in reduced Cl^−^ uptake and hydration of airway fluids [141], which renders mucosal cilia incapable of blocking foreign bodies, such as bacteria, and results in the failure of neutrophilic inflammation to abate and damage the respiratory system. Bacterial colonization in the lungs of CF patients and persistent neutrophilic inflammation suggest a defect in phagocytosis by lung macrophages, but the exact mechanism is not clear; the only understood mechanism is that phagocytosis of apoptotic cells by airway macrophages is impaired by the cleavage of the PS receptor by neutrophil elastase [142].

The pathogenesis of idiopathic pulmonary fibrosis (IPF) is still unknown, and its main features are alveolar structural disorders as well as diffuse pulmonary fibrosis, leading to the obstruction of gas exchange in the lungs, which ultimately leads to respiratory failure and death, and the disease is characterized by rapid progression, high morbidity and mortality and poor prognosis. IPF is often accompanied by chronic neutrophilic inflammation, and it has been experimentally demonstrated that the phagocytosis of apoptotic cells in bronchoalveolar lavage fluid (BALF) is reduced in patients with IPF [143,144]. The drugs currently used to treat IPF are mainly pirfenidone and nintedanib, and no clinical drug has been found to be able to play a therapeutic role by increasing macrophage efferocytosis, and inducing the expression of PPAR γ in airway macrophages may be a potential therapeutic pathway [82,145].

The phagocytic errors of macrophages have been found in all three of the above respiratory diseases where the pathogenesis is not yet known; thus, targeting efferocytosis to develop drugs has a promising future.

## 5. Current Drugs Targeting Efferocytosis

Defective efferocytosis has been identified in a number of different types of diseases, and therefore developing drugs targeting efferocytosis may be a promising therapeutic direction for the future. The drugs that have been shown to be effective against efferocytosis to date are described below.

The most prominent drugs capable of enhancing phagocytosis are glucocorticoids, which can exert this effect through a variety of mechanisms: (i) enhancement of downstream signaling through Mer and Pro S, (ii) alteration of the Rac/RhoA balance and (iii) modulation of PPAR γ-induced macrophage reprogramming [6]. Dexamethasone-treated macrophages have increased Rac expression, altered cytoskeletal structure and increased cell motility, all of which are required to carry out effective efferocytosis [24,146].

Statins are 3-hydroxy-3-methylglutaryl coenzyme A (HMG-CoA) reductase inhibitors that can help enhance efferocytosis by inhibiting the isoprenylation of RhoA [22,147]. Macrolide antibiotics may alter macrophage programming and increase the expression of bridging molecules to enhance uptake of apoptotic cells [6]. β-Hydroxybutyric acid, a dietary supplement, can increase the expression level of SLC2a1 in the presence of impaired sustained efferocytosis in phagocytes and the activity of the lysosomal arginine transporter protein Pqlc2, which can drive sustained phagocytosis in vitro and in vivo [5,77,148].

## 6. Conclusions

In recent years, the role of efferocytosis in the occurrence and development of diseases and the restoration of body homeostasis has been increasingly recognized. The clearance of apoptotic cells by phagocytic cells is a multi-step and interconnected process, and errors at any stage of the entire process may lead to inflammatory or autoimmune diseases. It has been proven that there is phagocytosis error by macrophages in respiratory system diseases, whether it is in chronic airway diseases such as COPD and asthma, which have been extensively studied, or in lung diseases such as ARDS and CF, whose pathogenesis is still unknown.

This review focuses on the role of efferocytosis, introducing its mechanisms, related respiratory diseases and targeted treatment methods. We focus on promoting inflammation resolution in respiratory diseases by improving the damaged efferocytosis. At present, with the efforts of researchers, some drugs that can act at different stages of the efferocytosis process to solve inflammation and treat diseases have been discovered, and they have achieved good therapeutic effects. Nevertheless, our understanding of the role of efferocytosis is still not profound enough. Whether there are any other “eat-me” signals, “don’t eat-me” signals, phagocytic receptors, anti-phagocytic receptors and bridging molecules; whether there are other signaling pathways that can be triggered after the recognition of apoptotic cells by the phagocytes; whether there is a synergistic effect between these signaling pathways; and more questions which all warrant further exploration. We firmly believe that the more thorough research is conducted on the role of the efferocytosis in the future, the more likely it is to provide more targets for the treatment of respiratory system diseases, which can develop more effective clinical drugs and provide more innovative and effective treatment strategies for respiratory system diseases and even other system diseases.

## Figures and Tables

**Figure 1 ijms-24-14871-f001:**
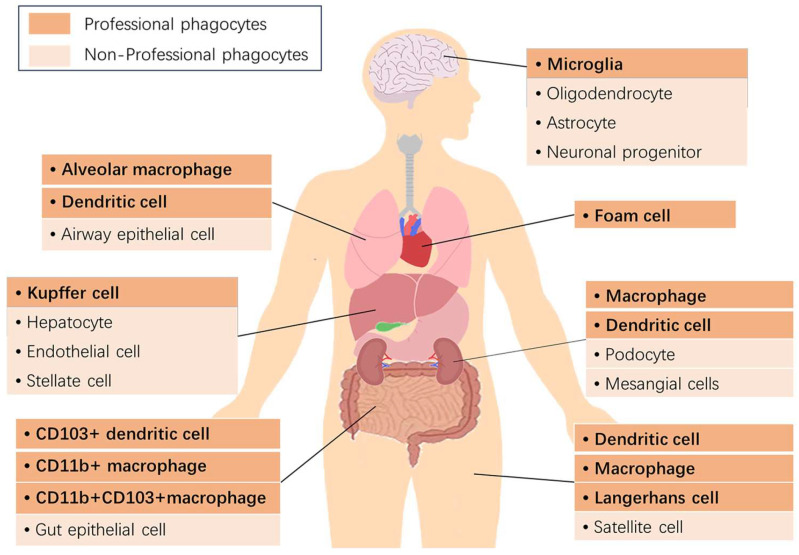
The professional phagocytes and non-professional phagocytes in the human body.

**Figure 2 ijms-24-14871-f002:**
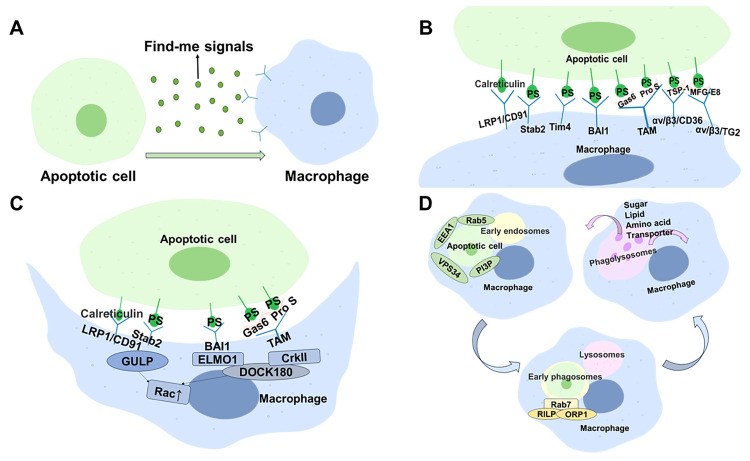
The mechanisms of efferocytosis. Take macrophages as an example. (**A**) The smell stage: apoptotic cells release “find-me” signals to recruit macrophages to the site of death. (**B**) The taste stage: Macrophages recognize apoptotic cells via “eat-me” signals, which are mainly phosphatidylserine (PS). This process occurs in two ways: one is direct binding of macrophage surface receptors to PS, such as BAI-1, Tim4 and Stabilin2, and the other is indirect binding to PS in the presence of the bridging molecules. Calreticulin is also an eat-me signal that can directly bind to LRP1/CD91 on the surface of macrophages. (**C**) The phagocytic stage: After phagocytes recognize apoptotic cells, Rac on the plasma membrane is activated and the cytoskeleton is rearranged under the plasma membrane, eventually forming phagosomes. There are two main pathways in this process. One is the recruitment of the articulin GULP after binding of LRP1 (CD91) to calreticulin or Stab2 to PS, which activates Rac. The other is the recruitment of the articulin CrkII to the cell membrane after indirect binding of MerTK to PS. Then, the articulin CrkII binds to the ELMO1-DOCK180 complex that is formed and activated by the binding of BAI-1 to PS. Finally, the CrkII-DOCK180-ELMO complex activates the RHO family, in particular, Rac. (**D**) The digestion stage: Phagosomes fuse with early endosomes in the presence of activated Rab5, VPS34 and PI3P to form early phagosomes. The transformation of early phagosomes into late phagosomes is marked by the absence of Rab5 as well as the acquisition of Rab7. Late phagosomes in turn fuse with lysosomes to form phagolysosomes under the synergistic action of activated Rab7, RILP and ORP1. The newly formed phagolysosome relies on proteases and other enzymes within it to digest the phagocytosed cell corpses, and the digested products are reused by macrophages or excreted from the body.

**Table 1 ijms-24-14871-t001:** Signals involved in the efferocytosis process.

	Signal	Receptor	Function
Find-me	ATP/UTP	P2Y2 [9,10]	Bind to receptors on phagocytes and recruit phagocytes to apoptotic sites
	LPC	G2A [11,12,13]	
	S1P	S1P1-5 [11,12,13]	
	CX3CL1	CX3CR1 [14]	
	ICAM-3	CD14 [18]	
	TSP-1	CD36 [18]	
Eat-me	PS	BAI-1 [19]	Directly integrate with PS, raise and activate ELMO1-DOCK180 complex
		TIM family [10]	Directly integrate with PS, TIM-4 triggers LC-3-associated phagocytosis
		Stabilin1/2 [20]	Directly integrate with PS, activate Rac via GULP
		TAM family(Tyr3, Axl, MerTK) [10,21]	Indirectly bind to PS, then dimerization occurs or recruit adapter protein CrkII, form ELMO1-DOCK180-CrkII complex, finally activate Rac
		αv/β3/CD36 [22]	Indirectly bind to PS via TSP-1
		αv/β3/TG2 [22]	Indirectly bind to PS via MFG-E8
	Calreticulin	LRP-1/CD91 [20,23,24]	Recruit adaptor protein GULP, activate Rac
Don’t eat-me	CD47	SIRP α [5]	Activate SIRPα and make it phosphorylated, activate phosphatase to inhibit phagocytosis
	CD24	Siglec 10 [5]	Highly express on tumor cells, inhibit phagocytosis
	CD31	[5]	Inhibit phagocytosis due to the homologous interactions between healthy cells and macrophages

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
