# Peer review of "Efferocytosis and Respiratory Disease"

_ijms, 2023, doi:10.3390/ijms241914871_

Round 1

Reviewer 1 Report

Zheng and colleagues present a review of the literature regarding the process and steps of efferocytosis, approaches for modulating efferocytosis, and the role of efferocytosis in lung diseases. Overall, the review is comprehensive, citing 146 published articles. Key molecular players are summarized in Table 1. Most of the suggested edits involve typographical or grammatical errors. There are a few instances where statements are made without citing appropriate references.  The authors cite several review articles but should make a strong effort to cite primary articles and not review articles.

A list of suggested, minor edits in order of appearance in the article follows:

1. Abstract line 13. Insert a space between sentences: “…of respiratory diseases.  In this…”

2. Introduction line 38: delete extra character:  “…involved in efferocytosisl” should be “…involved in efferocytosis”

3. Fig 2 legend line 57: “The macrophages stage” should be changed to “The phagocytic stage” to match language used in the first paragraph of Section 2.

4. Fig 2 legend lines 59-64:  This overly long sentence should be split into 2 to 3 separate sentences. In particular, the sentence fragment appearing on lines 62-64 should be split into two sentences.

5. Section 2.1, line 79: delete “,etc”

6. Figure 2 would be improved in the font sizes of the labels can be enlarged.

7. Section 2.2, line 103: consider changing “exposure to calreticulin” to “exposure of calreticulin”

8. Line 114: change “Apoptosis cells” to “Apoptotic cells”

9. Lines 117-118: the phrase, “…, and then the CD91 (low-density lipoprotein receptor protein) of calreticulin is cleared by phagocytes..” is unclear.  Consider changing to, “ and then the CD91 (low-density lipoprotein receptor protein) mediates calreticulin recognition and subsequent phagocytosis”

10. Line 150: change “eating me” to “eat me”

11. Line 153: change “phagocytosis” to “phagocytosed”

12. Line 154: there appears to be missing words. Consider changing, “…, the signal is superior…” to , “…, the “don’t eat me” signal is superior…”

13. Lines 182-183:  What is the reference for this statement?

14. Line 199: change, “Early phagosomes continues…” to “Early phagosomes continue…”

15. Lines 225-226:  What is the reference for this statement?

16. Lines 232-233: Please define the abbreviation on the first use in the manuscript. The correct way to define an abbreviation is to spell it out first, then place the abbreviation in parentheses. Thereafter, the abbreviation can be used. E.g. , “soluble receptor of advanced glycan end products (RAGE)”.

17. Line 252:  The phrase, “most of which are modified lipids” appears out of place and should be deleted.

18. Line 329: “Patrick F.” should be changed to “Asare, et al and others” or something similar.

19. Line 358: Asthma subheading is spelled incorrectly.

20. Lines 378 and 379:  IL-10 and TGF beta are incorrectly referred to as inflammatory mediators. These two cytokines in particular are more anti-inflammatory than pro-inflammatory.

21. Lines 443-444, 458: Please be sure to define acronymns upon the first use in the article, including MBL, IAV and MLE

22. Lines 529 and 531:  consider changing one or more uses of “very important” to “important” or “critical”

23. Line 543: insert a space between “the” and “published”

Minor typographical and grammatical errors are present.

Author Response

Please check my reply in the attachment.

Reviewer 2 Report

The idea of the review by which the authors are targeting the respiratory diseases is good, however reading through the MS was not that pleasant. The following concerns and comment I have:

OVERALL 

The English part is sometimes really inappropriate, thus the understanding of the MS is hard. 

The MS has many typos, mistakes. 

90% of the referred articles' numbers are not formatted properply. There is always a missing space between the end of the sentence and the reference number in brickets. 

ABSTRACT

Line 11: "If the process goes wrong, it will damage the tissue structure and function of the organism, and then cause different types of diseases." This sentences just does not any sense process related damage can not cause , just might lead to different diseases, or the development of the diseases. 

Line 14: "mechanims of macrophages efferocytosis." Again, it can have a meaning the efferocytosis of the macrophages, or the one which generated by the marcophages. 

INTRODUCTION
Line 23: The first sentence is non relatable, non scientific, "filling the gap" type of statement without any information. As the MS is a review it needs to be sharp and targeted. 

FIGURE 1. Instead of Organism It would better to use "human" as the figure shows a human body, and not all of the investigated genes and parameters were published in humans. 

Line 49: "The following will provide a detailed description..." the sentence it is not relating properly/comprehensive to the text and the line, even more it is not scientific.

Figure 2. Legends and decription, it was poorly edited like "example: A: " and many other mistakes were there which made the reading hard. The labels in the figure was impossible to read, the whole figure needed to be done more clear and visible.

Table 1. Abbreviations and meaning under the table. It looks like it was just placed there without anything, it was really confusing, there was no logic in it. 

Line 125 Tissue glutamine transferase is not the proper name of the current TG2 it can be labelled as tissue transglutaminase , or transglutaminase 2 etc. 

Line 150 " is to better phagocytize" typo

Many abbreviations are missing like: HODE, RAGE, 15-HETE, MerTK, NLRP3

Line 225 "In cancer, reducing the number of anti-phagocytosis receptors allows cancer cells to be engulfed, which may be a potential treatment." It is more about a target rather than a treatment. 

4.2 Asthama - huge typo

In vitro and in vivo should be in italic.

Conclusion

I do believe that this is the weakest part of the whole MS, thus it ends up in a repetition of the suspicious statements. rather than a comprehensive picture about the elevated field of science. 
I do miss the conclusion, I was reading many different parameters, genes, proteins scrambling together one by one, but I do not see the big picture. It left me unsatisfied. 

Also it was a big dissapointment that although the title said respiratory diseases, I hardly find any relationship with the information stated in the MS. 

The MS should be checked by a native speaker/scientific person, due to the inappropriate term, phrases. Some of the parts sounds really weird and inappropriate.

Author Response

(The authors gave the same response as above.)

Round 2

Reviewer 2 Report

The authors corrected most of the required modifications/comments.